# Resistance, Recovery and Resilience of Two Co-Occurring Palaeotropical *Pinus* Species Differing in the Sizes of Their Distribution Areas

**Le T. Ho** [1,2,*] **, Jana Hoppe** [1] **and Frank M. Thomas** [1]

1   Faculty of Regional and Environmental Sciences, Geobotany, University of Trier, Behringstraße 21,
    54296 Trier, Germany; s6jahopp@uni-trier.de (J.H.); thomasf@uni-trier.de (F.M.T.)
2   Department of Forest Resources Management, Faculty of Forestry, Nong Lam University HCMC,
    Thu Duc District, Ho Chi Minh City 721400, Vietnam
*   Correspondence: s6lehooo@uni-trier.de

**Abstract:** Using a dendrochronological approach, we determined the resistance, recovery and resilience of the radial stem increment towards episodes of growth decline, and the accompanying variation of $^{13}C$ discrimination against atmospheric $CO_2$ ($\Delta^{13}C$) in tree rings of two palaeotropical pine species. These species co-occur in the mountain ranges of south–central Vietnam (1500–1600 m a.s.l.), but differ largely in their areas of distribution (*Pinus kesiya* from northeast India to the Philippines; *P. dalatensis* only in south and central Vietnam and in some isolated populations in Laos). For *P. dalatensis*, a robust growth chronology covering the past 290 years could be set up for the first time in the study region. For *P. kesiya*, the 140-year chronology constructed was the longest that could be established to date in that region for this species. In the first 40 years of the trees' lives, the stem diameter increment was significantly larger in *P. kesiya*, but levelled off and even decreased after 100 years, whereas *P. dalatensis* exhibited a continuous growth up to an age of almost 300 years. Tree-ring growth of *P. kesiya* was negatively related to temperature in the wet months and season of the current year and in October (humid transition period) of the preceding year and to precipitation in August (monsoon season), but positively to precipitation in December (dry season) of the current year. The *P. dalatensis* chronologies exhibited no significant correlation with temperature or precipitation. Negative correlations between BAI and $\Delta^{13}C$ indicate a lack of growth impairment by drought in both species. Regression analyses revealed a lower resilience of *P. dalatensis* upon episodes of growth decline compared to *P. kesiya*, but, contrary to our hypothesis, mean values of the three sensitivity parameters did not differ significantly between these species. Nevertheless, the vigorous growth of *P. kesiya*, which does not fall behind that of *P. dalatensis* even at the margin of its distribution area under below-optimum edaphic conditions, is indicative of a relatively high plasticity of this species towards environmental factors compared to *P. dalatensis*, which, in tendency, is less resilient upon environmental stress even in the "core" region of its occurrence.

**Keywords:** basal area increment; paleotropis; pointer year; stable carbon isotope; tree-ring analysis; weather extreme

## 1. Introduction

*Pinus kesiya* Royle ex Gordon and *P. dalatensis* Ferré are tropical pine species that co-occur at intermediate elevations in the mountain ranges of south–central Vietnam but differ largely in the sizes of their distribution areas. *Pinus kesiya* is widely distributed from northeast India to the Philippines. This wide distribution is due to its adaptability to stressful environmental conditions including acidic, nutrient-poor soils, low water availability and exposure to wildfire [1,2], although it grows best on relatively humid, well-drained soils with an adequate nutrient supply [3]. Due to its high adaptability and relatively fast growth, especially in its early life stages, this species is also being used

for nature-oriented reforestation even at relatively dry and acidic sites [2]. It is a light-demanding species [3] that typically grows in open forests [1]. In Vietnam, this species occurs across the country, typically in drier montane areas [4], with the centres of occurrence in Lam Dong Province of Vietnam's southern central part and in Ha Giang Province in the north [2].

In contrast, the occurrence of the only rarely studied *P. dalatensis* (syn. *P. wallichiana* var. *dalatensis* (Ferré) Silba [5]) in Vietnam is mostly restricted to higher elevations (1400–2600 m a.s.l.) from the Dalat Plateau in Lam Dong Province in the south to Thua Thien Hue Province in the central highlands, usually in small populations of up to 100 trees [6], and to a small isolated population growing at 740 m a.s.l. in Gia Lai Province of south–central Vietnam [7]. Generally, *P. dalatensis* is associated with other coniferous tree species such as *Fokienia hodginsii* (Dunn) A.Henry & H.H.Thomas. and *Pinus krempfii* [8]. Outside Vietnam, some stands only grow at a relatively low elevation (800–1100 m a.s.l.) in the Nakai Nam Theu conservation area of Laos [6]. Due to its limited and fragmented distribution area and small population sizes, *P. dalatensis* is classified as "near threatened" according to IUCN nomenclature [9] and as "rare" with a proposition as "vulnerable" on a national level [10].

Resistance against and recovery and resilience upon stress episodes are decisive features for the persistence of plant species in their habitat and their ability to expand their distribution range. In the recent past, the resistance, recovery and resilience of plant species, and of woody plants in particular, have even gained importance against the background of the current climate change (e.g., [11–14]). In the seminal paper by Lloret et al. [15], these sensitivity parameters are calculated as the ratios of the tree growth before, during and after a given stress episode. This concept has also been successfully used to characterise the response of different coniferous *(Abies alba* Mill., *Juniperus oxycedrus* L., *J. phoenicea* L., *J. thurifera* L., *Picea abies* (L.) H. Karst, *Pinus nigra* J.F.Arnold, *P. sylvestris* L., *Pseudotsuga menziesii* (Mirbel) Franco*)* as well as broadleaf evergreen *(Quercus ilex* L.*)* and deciduous tree species *(Alnus glutinosa* (L.) Gaertn., *Fagus sylvatica* L., *Fraxinus angustifolia* Vahl, *Larix decidua* Mill., *Q. faginea* Lam., *Q. petraea* (Matt.) Liebl., *Q. pyrenaica* Willd., *Q. robur* L.*)* to drought episodes within and across biomes (e.g., [12,16–23]).

*Pinus kesiya's* broader range of distribution and habitats requires a relatively broad range of adaptation and acclimatisation ability, especially with regard to climatic conditions. Therefore, we hypothesise that *P. kesiya*, compared to *P. dalatensis*, exhibits higher resistance, recovery and resilience towards episodes of growth reduction, which may have been caused by weather extremes. We tested this hypothesis by growth analyses and analyses of stable carbon isotope discrimination against atmospheric $CO_2$ ($\Delta^{13}C$) as a drought stress indicator in stem increment cores derived from an elevation belt extending from 1500 to 1600 m a.s.l. in the Bidoup Nui Ba National Park of south–central Vietnam, where stands of both species grow in close proximity, and related the stem increment to weather data. Due to its protection status, the national park is an ideal site for conducting that kind of study. We also expect that the results will increase the knowledge on the ecology of the rarely studied *P. dalatensis* and contribute to understanding the cause of the different distributions of the two palaeotropical pine species.

## 2. Materials and Methods

### 2.1. Study Sites

The study was carried out in the Bidoup-Nui Ba National Park in the eastern part of the Dalat Plateau, central highlands of Vietnam, Lam Dong Province, in stands of the pine species that are typical of the region. The average minimum and maximum temperatures are 16 °C and 25 °C, respectively; the annual rainfall is 1500–2200 mm. At elevations above 800 m in this region, sub-tropical mixed broadleaf and coniferous forests dominate, which are characterised by rich and diversified needle-leaved species, including some conifers such as *Pinus dalatensis* (PIDA), *P. krempfii*, *P. kesiya* var. *langbianensis*, *Fokienia hodginsii* and *Taxus wallichiana* as well as some podocarps such as *Dacrydium elatum, Dacrycarpus*

*imbricatus* and *Podocarpus neriifolius*. The typical secondary *P. kesiya* (PIKE) forests dominate the dry area in research region.

We conducted the field sampling in March and April 2014 and 2015. Samples were collected from protected forests located between N 12°10′ and 12°12′ latitude and E 108°39′ and 108°42′ longitude at elevations between 1500 and 1600 m a.s.l. (Table 1). The research stands were located within an area of 12 km². The smallest distance among them was 100 m, and the greatest, 2 km. Study plots of 1000 m² (25 m × 40 m) that were representative of the forest were selected. The stand structural characteristics differed between the monospecific PIKE forest and the old-growth PIDA forest. The former was typically dominated by *P. kesiya* in the whole stand with few understory species. On that plot, prescribed burning of the litter was applied annually according to national park rangers' common practice to foster tree regeneration. This practice resulted in a litter layer of only a few centimetres and in the occurrence of charcoal particles in the Ah horizon of the soil. In the PIDA stands, *P. dalatensis* was co-dominant with other conifers in the canopy layer, including *P. krempfii* and some *Podocarpus* species. The litter layer was as thick as 30 cm, resulting in a deep Ah horizon of the soil. All stands shared the same soil type (shallow (ferric) cambisol on Cretaceous basaltic andesite and quartz–diorite–granite [24]). We used a Vertex IV ultrasonic hypsometer to measure the tree height and determined the diameter at breast height (dbh) with a diameter tape. On each plot, we sampled two stem increment cores per tree with a 5 mm diameter Haglöf borer from 6–13 healthy trees, and additionally from 3–4 trees of PIKE that grew at a distance of 20 m outside the plots to increase the reliability of the dendrochronological data set. More than 45 large dominant canopy PIDA trees (>120 years old) of varying diameters were sampled. However, the poor quality of some cores and age differences among PIDA trees resulted in significant variation in their growth pattern, leading to the rejection of several PIDA core samples in building up the master chronology. Therefore, the numbers of trees per plot that were precisely cross-dated and used for further analyses were 5, 5, 7 and 6 trees on the PIDA₁, PIDA₂, PIDA₃ and PIDA₄ plot, respectively (Table 1).

**Table 1.** Site and stand description of the study plots with different *Pinus* species. Tree density refers to all the particular *Pinus* trees growing on 1 ha. DBH, stem diameter at breast height. *PIKE, Pinus kesiya; PIDA, Pinus dalatensis*.

| Species | Elevation (m a.s.l.) | Longitude; Latitude | dbh (cm) | Tree Height (m) | Tree Density (Tree ha⁻¹) | Number of Analysed Trees |
|---|---|---|---|---|---|---|
| PIKE | 1487 | N 12°10.531′ E 108°39.863′ | 40.3 ± 2.2 | 24.8 ± 1.0 | 290 | 16 |
| PIDA₁ | 1494 | N 12°11.049′ E 108°41.526′ | 60.3 ± 0.5 | 30.0 ± 1.0 | 60 | 5 |
| PIDA₂ | 1525 | N 12°10.885′ E 108°40.394′ | 54.7 ± 2.0 | 24.3 ± 1.3 | 70 | 5 |
| PIDA₃ | 1535 | N 12°11.453′ E 108°40.762′ | 44.1 ± 1.2 | 23.8 ± 1.3 | 90 | 7 |
| PIDA₄ | 1546 | N 12°12.094′ E 108°40.642′ | 57.4 ± 2.1 | 24.9 ± 0.9 | 70 | 6 |

Cambial growth of the trees starts at the end of the dry period in February and ceases during the dry period from December through January [25]. Thus, annual rings can be identified in both species.

All further analyses were conducted at the Geobotany Department of Trier University.

### 2.2. Tree-Ring Analyses

The cores were prepared in accordance with standard dendrochronological methods [26,27]. Wood samples were air-dried, glued on a wooden mould and the uppermost layer was cut using a microtome (Swiss Federal Research Institute WSL, Birmensdorf, Switzerland) until the tree rings were visible. The two pine species showed a definite

annual ring structure, and cross-dating was readily established. Tree-ring widths were visually measured with 0.001 mm accuracy using a binocular microscope and a Lintab 5 device (Rinntech, Heidelberg, Germany) linked to a computer. The measured series was then statistically cross-dated with TSAP-Win Scientific software (version 4.64, Rinntech, Heidelberg, Germany), which tests tree-ring series against each other by correlation, g-score (co-linearity or 'Gleichläufigkeit' [28]) and cross-date index (CDI) [27]. After cross-dating, regional curve standardisation (RCS) [29], an age-detrended method, was applied on individual tree-ring chronologies using the program ARSTAN [30], followed by a robust mean to reduce the influence of outliers in the computation of the mean chronology. The inter-series correlation (Rbar) and the expressed population signal (EPS) were used to evaluate each chronology's reliability [31]. Both Rbar and EPS were computed over a 40-year window with a 20-year lag. The chronology's most reliable period was determined by a threshold of EPS at 0.85 [31]. To test for weather effects on tree growth, master chronologies from 23 trees of all four PIDA plots and from 16 trees of one PIKE plot were created.

The basal area increment (BAI), which provides a biologically meaningful variable showing growth trends, is calculated from tree ring-width series assuming stem growth is approximately concentric [32] by the following formula:

$$\text{BAI} = \pi \left( r_t^2 - r_{t-1}^2 \right) \tag{1}$$

where $r_t$ is the stem radius at the end of the annual increment $t$, and $r_{t-1}$ is the stem radius at the beginning of the annual increment $t$.

Since during the first 40 years of cambial age, both species' growth exhibited a linear growth trend, but with a different pace, the stem increments during this period were investigated separately.

### 2.3. Cellulose Extraction and Stable Carbon Isotope Analysis

Five tree cores (one per tree) with clear boundaries and no missing rings, from trees that were also included in the respective stand chronologies, were selected from each species. From these cores, all the rings formed from 1901 to 2013 were used for stable carbon isotope analyses.

The $\alpha$-cellulose was extracted from an individual tree-ring sample, which was cut into tiny pieces and enclosed in Teflon filter bags marked with a binary code [33]. These bags were then incubated in a 5% sodium hydroxide (NaOH) solution and heated in a water bath at 60 °C for 2 h. To these bags, a 7% sodium chlorite (NaClO$_2$) solution, which was brought to pH 4–5 with 96% acetic acid, was added and the 10 h incubation treatment was repeated three times at the same temperature. These steps were for removing fats, resins, oils, tannins and some hemicelluloses from the samples. The IAEA standard wood was used to determine the stable carbon isotope ratios ($\delta^{13}$C) from cellulose using an isotope ratio mass spectrometer (DeltaV Advantage, Thermo Fisher Scientific, Waltham, MA, USA) at a precision of 0.05‰.

The tree ring carbon isotope ratio of cellulose is quoted as $\delta^{13}$C values,

$$\delta^{13}\text{C} = (\text{R}_{\text{sample}}/\text{R}_{\text{standard}} - 1) \times 1000‰ \tag{2}$$

where R = $^{13}$C/$^{12}$C and R$_{\text{sample}}$ and R$_{\text{standard}}$ are the isotope ratios measured in the sample and the standard.

A large amount of isotopically "light" carbon has been emitted into the atmosphere due to the use of fossil fuels since the onset of the industrial revolution. This was sufficient to decrease the $\delta^{13}$C values in atmospheric CO$_2$ and in plants. Therefore, to render tree-ring $\delta^{13}$C records comparable across decades, it is necessary to filter out the anthropogenic imprint of fossil fuels [34,35]. This correction was done by calculating the $^{13}$C discrimination

against the atmospheric $CO_2$ composition ($\Delta^{13}C$) according to McCarroll and Loader [35] and Saurer and Siegwolf [36].

$$\Delta^{13}C = \frac{\delta^{13}C_{atm} - \delta^{13}C}{1 + \delta^{13}C/1000} \ (‰) \tag{3}$$

where $\delta^{13}C$ is the isotope signature of a tree ring and $\delta^{13}C_{atm}$ is the isotopic signature of the respective year's atmospheric $CO_2$.

The common signal among samples was determined according to customary dendrochronology procedure with Rbar statistic and EPS [31], both established over the period from 1901 to 2013.

### 2.4. Climate Data

As no climate station was installed in the national park, we obtained monthly data on mean temperature and precipitation for the 1901–2013 period from KNMI Climate Explorer [37], which provides quality-controlled data at a high spatial resolution. We compared pine chronologies with the $0.5° \times 0.5°$ grid CRU TS 4.02 data set of temperature produced by Harris et al. [38] and with the gridded precipitation series GPCC V2018 produced by Schneider et al. [39]. The selected grid was delimited by the following coordinates: 12.0–12.5° N and 108.0–108.5° E. We analysed the correlations between ring-width chronologies, $\Delta^{13}C$ ratios and climate variables. Since the radial growth of trees is usually also determined by the year's climate prior to ring formation, the climate variables were taken for the previous fifteen months back to October of the previous year until December of the current year. For the correlation analyses, the months were additionally grouped according to the seasonal climatic conditions: dry period, December–February (DJF); humid pre-monsoon period, March–May (MAM); wet monsoon season, June–August (JJA); humid transition period, September–November (SON).

### 2.5. Stress Events and Sensitivity Indicators

To single out the main factors affecting growth, we identified the years with the severest reductions in BAI. We determined these negative pointer years as those years in which at least 60% of a species' BAI series displaying a BAI decrease of at least 25% relative to the average BAI of the five preceding years. Thus, the following pointer years were identified: 1830, 1888, 1906, 1950, 2000–2001 for PIDA and 1977–1978, 2002–2003 for PIKE. Hereafter, we refer to each of these episodes according to the first years. For analysing the growth response to the selected episode of growth decline, we calculated the mean annual BAI in the five years before (*PreEps*) and after (*PostEps*) the event. We calculated resistance ($R_t$), recovery ($R_c$) and resilience ($R_s$) indices, according to Lloret et al. [15]. The indices were computed individually for each tree from its mean BAI series as follows:

$$\text{Resistance } R_t = Eps/PreEps \tag{4}$$

$$\text{Recovery } R_c = PostEps/Eps \tag{5}$$

$$\text{Resilience } R_s = PostEps/PreEps \tag{6}$$

Resistance ($R_t$) is considered as the capacity to sustain growth levels during the stress episode and represents the decrease from the pre-episode to the stress period. Recovery ($R_c$) is the ability to recover from stress during the disturbance. Resilience ($R_s$) is defined as the capacity to return to the growth level before the disturbance.

In addition, we calculated the relative resilience $rR_s$ according to Lloret et al. [15] as the resilience weighted by the injury experienced during disturbance:

$$\text{Relative resilience } rR_s = (PostEps - Eps)/PreEps \tag{7}$$

$$\text{which translates into } rR_s = R_s - R_t \tag{8}$$

Thus, the relative resilience is high when the impact exerted by the disturbance is relatively strong (low *Eps* due to low resistance), the recovery is high (high *PostEps*) or the growth before the disturbance is relatively low (low *PreEps*).

The four indices were calculated individually on the basis of the BAI ($cm^2$ $year^{-1}$) for 16 and 23 sample trees of PIKE and PIDA, respectively. Interspecific comparisons were made on the basis of the average sensitivity indicators computed for each species.

### 2.6. Soil Analyses

The uppermost 20 cm of the mineral soil were sampled as vertical mixtures after removal of the litter from five different positions within the plot and analysed separately for each stand. Before further analyses, charcoal particles resulting from prescribed burning were manually removed and sieved. Soil pH was measured in water ($pH_{H2O}$) and 0.1 N KCl ($pH_{KCl}$). The concentrations of total nitrogen (N) and carbon (C) were measured with an elemental analyser (Flash EA 1112, Thermo Fisher Scientific, Waltham, MA, USA). Plant-available phosphorus ($P_a$) was determined photometrically following the procedure of the molybdenum blue method in a calcium–acetate–lactate (CAL) extract [40]. Acidic cations, including exchangeable aluminium (Al), iron (Fe) and manganese (Mn), and neutral cations (calcium—Ca, magnesium—Mg, potassium—K, sodium—Na) were measured by atomic absorption spectrometry (contrAA 300, Analytik Jena, Jena, Germany) after acidic wet digestion.

### 2.7. Statistical Analyses

Data sets with means $\pm$ 1 standard error (SE) are presented if not indicated otherwise. Based on the 113-year annual mean, the BAI of the two species was compared using the non-parametric Mann–Whitney U test. Sensitivity indicators were compared between the species using the unpaired Student's *t*-test. Pearson correlations were computed between tree-ring width indices, $\Delta^{13}C$ ratios and climatic variables. For testing the difference among the stable carbon isotope signals of the two pine species, we used ANOVA for repeated measurements for the whole 10 time series, even if requirements of non-significant deviations from normality and from homogeneity of variances were not met. We tested the relationship between the four sensitivity indices $R_t$, $R_c$, $R_s$, $rR_s$ and the predictors PreEps, tree age, dbh at the level of individual trees with linear mixed-effects models:

$$S_i = \beta_0 + \beta_1 PreEps + \beta_2 age + \beta_3 dbh + \varepsilon_i \tag{9}$$

where $S_i$ represents the sensitivity indicator of tree *i*, $\beta_0 \ldots \beta_3$ are fixed effects associated with the tree-level covariates *PreEps*, *age* and *dbh*, and $\varepsilon_i$ is the group error vector. SPSS (version 25; IBM, Armonk, NY, USA) was used for conducting statistical tests with a significance level of $p < 0.05$ if not indicated otherwise.

## 3. Results

### 3.1. Stand Structure, Soil Conditions and Tree Growth

The relatively old PIDA trees exhibited large sizes with dbh from 44.1 cm to 60.3 cm, corresponding to tree heights from 23.8 m to 30.0 m (Table 1). Soil reactions (Table S1) were strongly acidic in the PIDA plots, but somewhat less acidic in the soil of the PIKE stand. High levels of Al concentration (>300 $mmol_c$ $kg^{-1}$) were found in the soil of PIKE and PIDA stands. Concomitantly, the values of base cation concentrations were very low. The concentrations of total N and $P_a$ were also low and exhibited slight variations among the plots, while the percentage of total C in the PIKE plot was higher than in all PIDA stands.

We were able to cross-date the annual growth rings and, for the first time, to construct a robust chronology for PIDA growth, covering the past 290 years in the research region. In the case of PIKE, the 140-year chronology was the longest that could be established in that region for this species (Figure 1). The PIKE ring widths showed low growth from 1908 to 1912, sandwiched by two periods of large increments from 1889–1902 and 1913–1922. The rest of the chronologies experienced variation in growth, including two low figures for

1971–1979 and 2002–2008. PIDA chronologies displayed periods of reduced as well as of enhanced growth throughout the past two centuries (Figure 2a,b). Most notable for the ring widths is the below-average growth in the 1800s–1830s, the 1870s and the period from the 1930s to the 1940s. Periods with high growth rates are not particularly visible except for the 1760s to 1770s, a period that, however, included only two trees of a young age. The period of rapid growth in the first phase of the trees' life cycle ranged over 60 years in PIKE, but extended to 150 years in PIDA (Figure 2a,b). During the early growth stage (first 40 years), PIKE trees grew at a faster pace than PIDA, which was evident from a significant difference between the slopes of the linear regressions of BAI vs. age ($F = 47.29$; $p < 0.001$; Figure 2c). After that stage, PIDA exhibited large fluctuations in growth with time, while PIKE grew at a stable pace before showing a declining trend after 125 years. Considering the growth of the species in the most recent 113 years, the mean BAI value of PIDA (15.48 cm$^2$ year$^{-1}$) was significantly larger than that of PIKE (11.9 cm$^2$ year$^{-1}$) ($p < 0.001$).

### 3.2. Tree-Ring Indices and Climate Signals

The mean segment lengths of these two pine chronologies are 140 years (PIKE) and 290 years (PIDA), respectively. Based on the 0.85 threshold of EPS statistics, the site chronologies met signal strength requirements after 1900 for PIKE (10–16 trees) and after 1820 for PIDA (16–22 trees) (Figure 1). Thus, the resulting regional chronology is capable of reflecting climatic variations of a multi-decadal length.

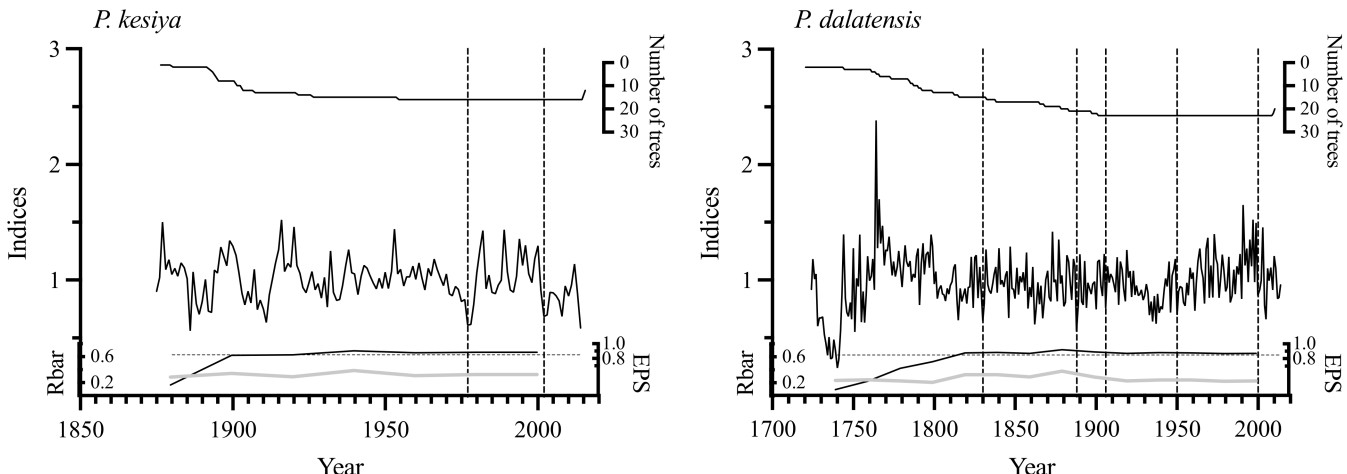

**Figure 1.** Chronologies of tree-ring indices of the two pine species. The *vertical dashed lines* indicate marker years (reduced growth). In the lower parts of the panels, the series of Rbar values are drawn in grey, and the series of EPS values, in black.

For PIKE, correlation analysis (Figure 3) revealed significantly negative relationships between the standardised tree-ring width and the temperature of the wet months and season (June–August), including the hottest months of the current year, and October (part of the humid post-monsoon period) of the preceding year. Growth was also negatively correlated with precipitation in August, but positively with precipitation in December (part of the dry period) of the current year. The correlations between the PIDA chronology on the one hand and temperature and precipitation on the other were weak for most months and statistically not significant.

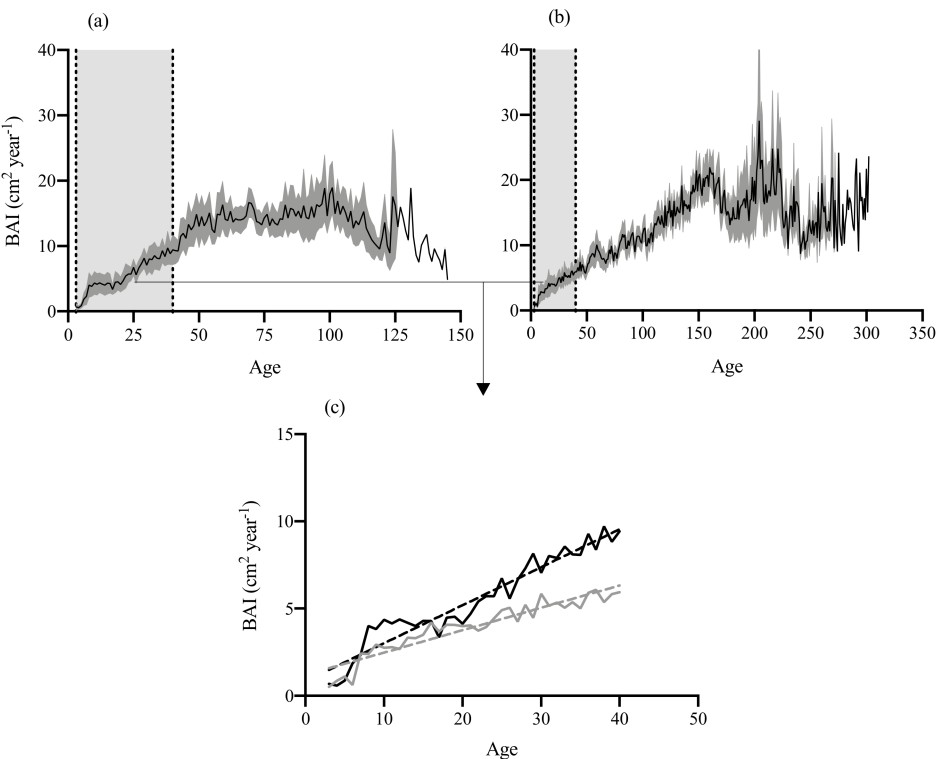

**Figure 2.** Course of mean basal area increment (BAI) against age. (**a**) *P. kesiya*; (**b**) *P. dalatensis*. *Vertical grey bars* mark the juvenile stage of the trees. Standard errors (SEs) are indicated by *grey areas*, means without SEs originated from 1 tree. (**c**) Correlation between the mean BAI of the species and age during their juvenile periods (age 3 to 40) with *dashed lines* indicating linear regressions: *black lines*, *P. kesiya* (BAI = 0.2180age + 0.8312); *grey lines*, *P. dalatensis* (BAI = 0.1283age + 1.194). The slopes of the two regression lines differ significantly.

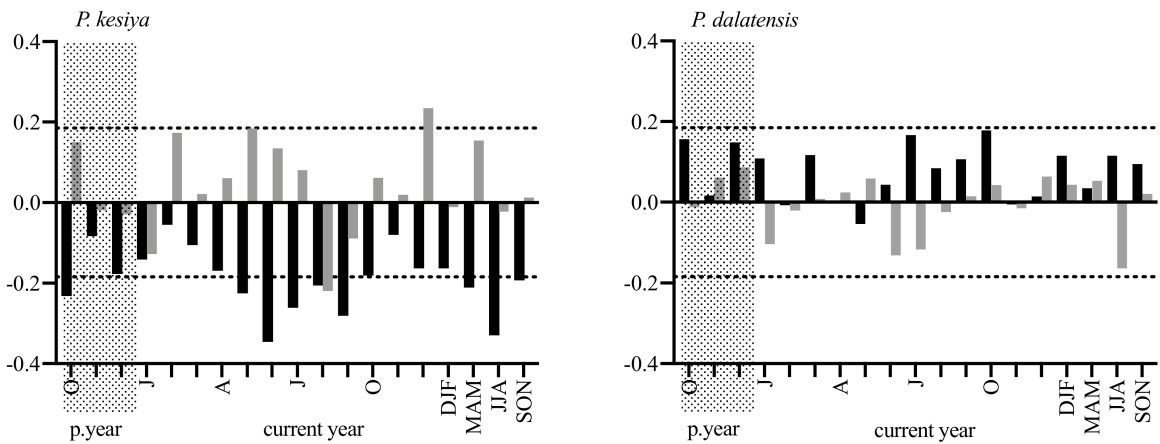

**Figure 3.** Correlation coefficients of tree-ring indices of the two pine species with *temperature* (*black bars*) and *precipitation* (*grey bars*) from 1901 to 2013. *Dotted lines* indicate thresholds of significance at $p < 0.05$. The letters on the *x*-axis indicate calendar months and seasonal periods. *DJF*: December (preceding year) to February; *MAM*: March to May; *JJA*: June to August; *SON*: September to November.

### 3.3. Sensitivity Indicators

In both species, there was no significant difference among individual trees. Therefore, a species-wise combination could be applied for further statistical analyses. A direct comparison of the sensitivity indicators between the species (Table 2) showed generally

higher $R_t$, $R_s$ and $rR_s$ in PIKE compared to PIDA, though these differences were not statistically significant.

**Table 2.** Sensitivity indicators ($R_t$—resistance, $R_c$—recovery, $R_s$—resilience and $rR_s$—relative resilience) of *P. kesiya* ($n$ = 32) and *P. dalatensis* ($n$ = 103).

| Species | Resistance ($R_t$) | Recovery ($R_c$) | Resilience ($R_s$) | Relative Resilience ($rR_s$) |
|---|---|---|---|---|
| *P. kesiya* | 0.70 ± 0.02 | 1.48 ± 0.08 | 1.03 ± 0.06 | 0.33 ± 0.05 |
| *P. dalatensis* | 0.67 ± 0.02 | 1.51 ± 0.06 | 0.96 ± 0.03 | 0.29 ± 0.03 |

Growth after low-growth episodes was positively correlated to the growth before these episodes in the two species (Figure 4). While in PIKE, the slope of that linear regression was not significantly different from the 1:1 ratio (F = 1.643; $p$ = 0.209), it was significantly less steep than the slope of 1:1 (F = 111.6; $p$ < 0.001) and the slope of the linear regression of PIKE (F = 47.29; $p$ < 0.001) in the case of PIDA, which is indicative of a lower resilience of this species. Respective relationships with regard to $R_t$, $R_c$ and $rR_s$ were not significantly different between both species.

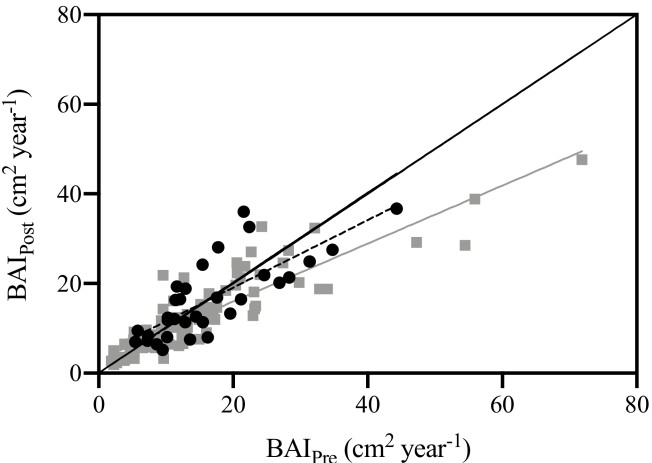

**Figure 4.** Relationship between BAI after and before drought episodes. *Black circles* and the *black dashed line* indicate the values and linear regression ($\text{BAI}_{\text{Post}}$ = 0.7529$\text{BAI}_{\text{Pre}}$ + 3.344) of *P. kesiya* ($n$ = 32); *grey squares* and the *grey solid line* indicate the values and linear regression ($\text{BAI}_{\text{Post}}$ = 0.6466$\text{BAI}_{\text{Pre}}$ + 3.025) of *P. dalatensis* ($n$ = 103). Solid black line, 1:1 ratio. Linear regression of PIDA (*grey solid line*) differs significantly from the others.

Recovery and resilience of tree growth were generally controlled by age and dbh (Table 3). In PIKE, the most decisive influence on $R_c$ and $R_s$ was exerted by age, followed by dbh, whereas in PIDA, dbh had a larger effect on $R_s$ than age. In general, the growth prior to episodes (PreEps) rarely influenced the sensitivity indicators, with the exception of a negative PreEps effect on the resilience ($R_s$ $rR_s$) of PIDA. Resistance ($R_t$) was the only parameter that was not correlated to any potentially predisposing factor.

**Table 3.** Correlation coefficients of parameters for fixed effects in a linear model relating sensitivity indicators of *P. kesiya* (*n* = 32) and *P. dalatensis* (*n* = 103) to pre-stress (PreEps) growth, diameter at breast height of stress episodes (dbh) and tree age (Age). The significance of the fixed effects is indicated with * for *p* < 0.05 and ** for *p* < 0.01.

| Species | Fixed Effects | $R_t$ | $R_c$ | $R_s$ | $rR_s$ |
|---|---|---|---|---|---|
| *P. kesiya* | PreEps | −0.09 | −0.16 | −0.16 | −0.15 |
| | Age | −0.17 | −0.64 ** | −0.64 ** | −0.69 ** |
| | dbh | −0.04 | −0.36 * | −0.32 * | −0.37 * |
| *P. dalatensis* | PreEps | −0.16 | −0.19 | −0.39 * | −0.30 * |
| | Age | 0.05 | −0.28 * | −0.23 * | −0.29 * |
| | dbh | −0.08 | −0.23 * | −0.31 * | −0.28 * |

*3.4. Stable Carbon Isotope Signals*

The cellulose $\delta^{13}C$ of PIKE ranged from −25.95‰ to −22.06‰ and averaged at −23.69‰. The PIDA $\delta^{13}C$ had a larger range, from −25.00‰ to −20.40‰, and averaged at −22.25‰, which is significantly higher than that of PIKE (*p* < 0.001). A continuously decreasing trend was observed for the PIDA $\delta^{13}C$ values from 1901 to 2013 with an annual decline rate of around 0.02‰ year$^{-1}$ (Figure S1).

The $\Delta^{13}C$ values of three out of five PIKE cores showed highly significant inter-series correlations over the time span, as determined by the Rbar (0.63) and EPS values (0.835), which, for EPS, was below the 0.85 threshold. For PIDA, we found relatively high Rbar (0.67) and EPS (0.91) values during the last 113 years in the $\Delta^{13}C$ data among the five cores. Thus, the $\Delta^{13}C$ series of both pine species yielded high Rbar and EPS values with a much smaller sample size in comparison with the number of cores used for the tree-ring chronologies. Therefore, the average time series can be considered to provide robust estimates of the mean tree-ring $\Delta^{13}C$ at the study sites. The $\Delta^{13}C$ data from PIKE exhibited a decrease across the investigated time period, whereas in PIDA, these data varied within a relatively small range without displaying a distinct trend (Figure 5).

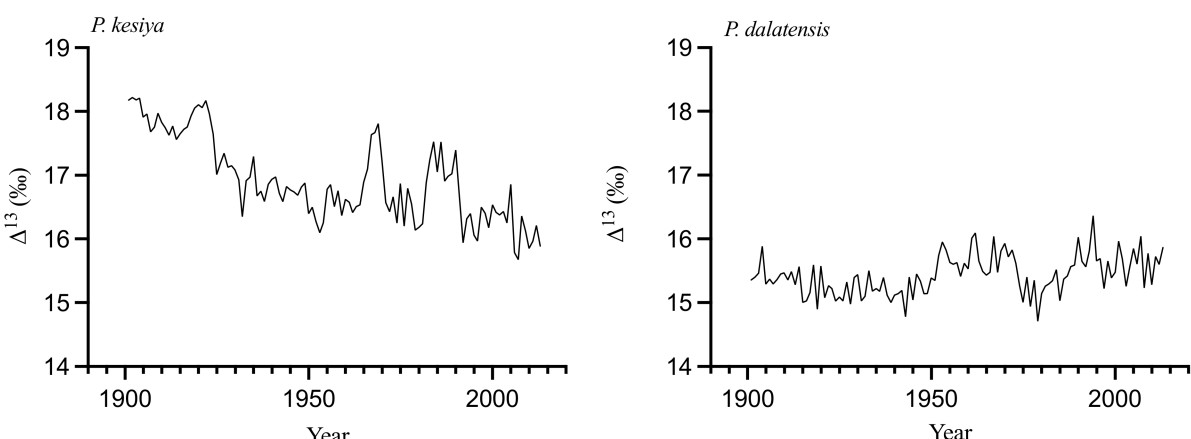

**Figure 5.** Chronologies of $^{13}C$ discrimination against atmospheric $CO_2$ ($\Delta^{13}C$) of *P. kesiya* (*n* = 3) and *P. dalatensis* (*n* = 5) from 1901 to 2013.

For PIKE, the correlation analysis (Figure S2) revealed significantly negative relationships between the $\Delta^{13}C$ chronology and the temperature across the wet season (June to October) until the beginning of the dry season (November to December) of the current and the preceding year. The correlation of the isotope signal was positive with the precipitation in December and the dry season (December to February). The correlations between the PIDA $\Delta^{13}C$ chronology and temperature were positive for most months and statistically significant in the monsoon season (June to August) and October in both the current and previous year. The isotope signal reflects the variation of precipitation in the whole year

rather weakly as the only significantly positive relation between the two variables was found in August.

In both species, the correlations between growth (BAI) and $\Delta^{13}C$ were weak, though significant; they were slightly tighter for PIDA ($r = -0.21$, $p < 0.001$) than for PIKE ($r = -0.19$, $p < 0.001$) (Figure 6). Since the $\Delta^{13}C$ values in the negative pointer years of *P. kesiya* (1977, 2002) and *P. dalatensis* (1906, 1950, 2000) were not lower than before or after the episodes of growth depression, it was not reasonable to calculate the sensitivity indicators for the isotope signatures.

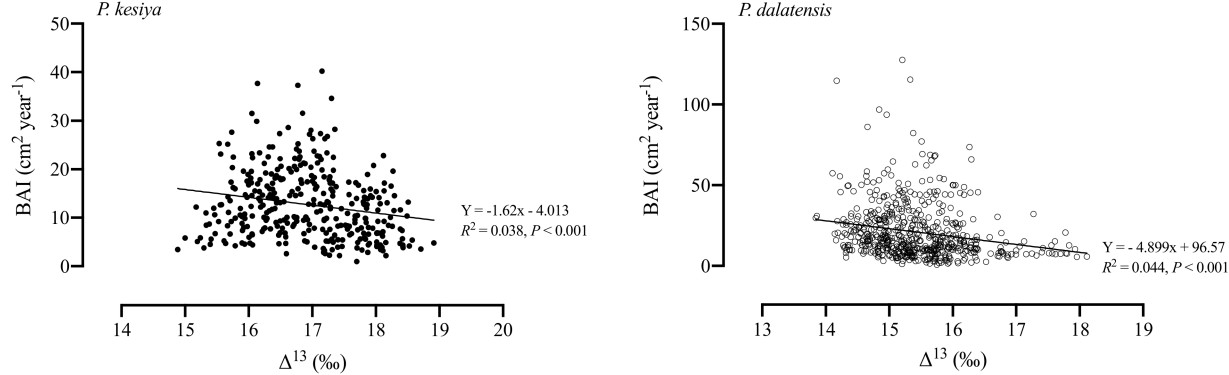

**Figure 6.** Correlation between the basal area increment (BAI; 1900–2013) of *P. kesiya* (*n* = 340), *P. dalatensis* (*n* = 575) and $^{13}C$ discrimination against atmospheric $CO_2$ ($\Delta^{13}C$) of the respective tree rings. *Solid line* indicates the regression.

## 4. Discussion

### 4.1. Tree Growth and Climate Signals

According to our expectation based on PIKE's generally faster growth in the early stage of its life, this species exhibited higher growth rates than PIDA during the first 40 years. This faster growth is also fostered by the relatively open canopy structure of the PIKE stands. In contrast, typical PIDA stands are relatively dense. However, *P. dalatensis* is not particularly shade tolerant and may require gaps to grow into the canopy after outlasting periods of shading during its juvenile stages. Upon canopy opening, the young trees can rapidly grow tall [2].

Although the radial stem increment of PIKE decreased after the age of approximately 100 years on our study plot, the trees were capable of reaching the threshold of their maximal age of 150 years [3]. This finding and the relatively fast growth at our study site during the first decades of life underpin the relatively high plasticity of this species as a precondition to grow well even in a part of its distribution area that, according to Luu and Thomas [2], is relatively isolated and only provides sub-optimal growth conditions. Those stands, therefore, can be considered as belonging to the marginal part of *P. kesiya*'s distribution area, where the edaphic conditions are not optimal for a species that generally requires sites with more or less humid soils with an adequate nutrient supply [3]. Hence, *P. kesiya* might represent one of the exceptions (see also [41]) of the often observed pattern that plant specimens growing towards the margins of their species' range occur in lower abundances and grow less well than their conspecifics in the centre of the species' distribution area [42,43], or are displaced to particularly favourable sites in marginal areas, as has been observed for several coniferous species (including one pine species) in North America [44]. On our study plot, the growth of PIKE in a climax-like pattern might have also been fostered by prescribed burning as a management tool to facilitate recruitment. This not only reduced competition by understorey species but also enhanced its growth by accelerating the release of nutrients from the litter. However, the linkage between the relatively high growth rates and the low maximum age of *P. kesiya* corresponds to the pattern that has been found in a similar manner for conifer species in western North

America, Spain, north and central Europe and Russia [45–47] and seems to be a general feature of trees (cf. [48]).

In contrast, PIDA grows in closed tropical forests, where trees are under more intense competition for light and nutrients. Upon the formation of gaps due to a tree fall or crown break, the trees can grow relatively fast during the early stage of their lives, but with a slower pace than the PIKE trees at our study sites. Once they have reached the canopy, they grow with large fluctuations according to the availability of resources, but can become several hundred years old [2]. In contrast to its growth trend during the early life stages, PIDA growth took the lead over PIKE in comparable periods during the mature stage (most recent 113 years). This result is congruent with findings of continued vigorous growth in large and old tropical and temperate tree species [49,50].

Our results of negative correlations of diameter increment with temperature in PIKE during the dry period in the hottest months of the year are in accordance with the results from studies of the same species conducted in northwestern Thailand [51] and northeastern India [52]. The negative correlation of tree growth with precipitation during the monsoon season in August, however, is in contrast to the results obtained by Pumijumnong and Wanyaphet [53] of a positive association of *P. kesiya*'s diameter increment with precipitation during the monsoon period in September. Our findings, however, confirm the results of a previous investigation on *P. kesiya* growth along an altitudinal gradient in the same region. In that study, growth of the species was negatively related to temperature in several periods of the year and positively to the precipitation of the pre-monsoon period (March to May) in six out of eleven altitudinal belts. However, tree growth was not immediately linked to the water supply across the entire elevational gradient [54].

In PIKE, the negative correlation between growth and precipitation in August, i.e., in the monsoon season, might be due to a more intense cloud cover that decreased solar irradiance and, thus, carbon gain and growth (cf. [55]), whereas, most probably, the positive correlation between precipitation and growth during the dry season in December has alleviated the restriction of water supply. Nevertheless, the growth of both pine species was only weakly associated with precipitation and the regression coefficients were not significant with only the above-stated exceptions for PIKE. This finding may also be explained by the fact that our study site hardly experienced a drought as severe as in other areas of Vietnam's central highlands. For instance, the El Niño–related droughts of 1997 and 2004 were widespread in the entire central highlands, resulting in reduced crop production from 1997–1998 and 2004–2005 [56]. However, neither the pine growth chronologies nor the climate data of our study region displayed a distinct relation to these events. In PIDA, no clear relationships could be found between growth and weather variables, and the ultimate cause of the detected episodes of growth decreases remains unclear for this species.

In general, humidity and temperature exert opposite effects on $\Delta^{13}C$: higher humidity results in larger isotopic discrimination (due to enhanced stomatal opening), and higher temperature, due to its common coincidence with increased vapour pressure deficit, to lower $\Delta^{13}C$ (upon stomatal closure). Accordingly, in PIKE, we found the expected positive correlations between $\Delta^{13}C$ and precipitation, and negative correlations between temperature and $\Delta^{13}C$, at least for several time periods in the off-monsoon season. In PIDA, the response of $\Delta^{13}C$ to these climate variables was much weaker. A possible reason might be the generally higher moisture supply at the PIDA site due to a more humid microclimate in the dense stands and the thicker litter layer that stores humidity. However, the negative correlations between BAI and $\Delta^{13}C$ and the lack of a decrease in $\Delta^{13}C$ in the episodes of growth decline indicate that in both species, growth was not impaired by drought in the elevational belt under investigation.

### 4.2. Growth Sensitivity Indicators

Contrary to our hypothesis, the averages of the sensitivity indices did not differ significantly between the two pine species. For PIKE, this finding fits our result on the

growth behaviour, demonstrating that this species is capable of thriving even at marginal sites of its range under sub-optimal edaphic conditions in contrast to PIDA, which, at our study sites, grows under typical conditions. Nevertheless, the less steep slope of the relationship of BAI after the episode of growth reduction ($BAI_{Post}$) vs. BAI before growth reduction ($BAI_{Pre}$; Figure 4) in PIDA is indicative of a trend in this species to a lower resilience compared to PIKE, despite the fact that the calculated average $R_s$ values did not differ significantly between the species. The larger number of pointer years in PIDA than in PIKE is another indication of a higher sensitivity of the former species towards disturbing events. In contrast to PIKE, relatively high growth rates before a stress episode decrease the relative resilience in PIDA, which is indicative of a relatively strong disturbance effect during the episode of reduced growth when the growth before the disturbance was lush.

For both species, in general, the growth performance changed during the trees' ontogeny. Young trees were overall more resilient and recovered better from low-growth periods (negative relationships between age on the one hand and $R_c$ and $R_s$ on the other). Higher tree age also had a negative effect on the recovery from a stress episode in *P. sylvestris*, a pine species widely distributed from Europe to the eastern parts of Central Asia [57]. Likewise, a negative correlation between tree size (dbh) and resistance to drought as has been found in our study was also detected in *P. sylvestris* of the Spanish Sierra Nevada mountain range [22]. The decrease in $R_c$ and $R_s$ with age (and, thus, tree size) might be explained by a decreasing capability of rapidly initiating repair mechanisms due to a reduced photosynthetic capacity [58].

In interpreting the results on the resilience of the species, it has to be considered that the sensitivity indicators used in the present and in previous studies are mathematically not independent from one other [15]. Moreover, as they are relative values of the growth ratios before, during and after stress episodes, they are strongly influenced by the growth prior to the stress event (cf. [59]). Thus, in theory, a high resistance index could just be effected by poor growth before the stress episode, preventing a strong growth decline during stress, and a corresponding argument is valid for the resilience index. Nevertheless, the resilience approach used here and in other studies proved to be useful in assessing the growth responses of trees to stress events in comparisons within or between species or even among entire forests ([59,60] and literature cited in the Introduction).

All in all, the tendency towards a higher resilience and lower susceptibility to disturbance events (smaller number of negative pointer years) in PIKE is another indication of a higher ecological plasticity of this species. Such increased plasticity might be (i) connected to the larger distribution range (cf. [61]), which was even larger than today at the time of the species' evolution in the late Miocene [62,63] and implies a larger variation in growth conditions among the populations, and (ii) facilitated by an extensive gene flow among *P. kesiya* and the closely related *P. (tabuliformis* var.*) densata* and *P. yunnanensis* [64].

## 5. Conclusions

According to our expectation, we found a higher growth increment in *P. kesiya* compared to *P. dalatensis* during the first decades of the species' life cycles although, at our study site, the former species grows at the margin of its distribution area under below-optimum edaphic conditions. We also detected more events of a significant decrease in growth (greater number of negative pointer years) and a tendency towards a lower resilience upon periods of growth decline in *P. dalatensis*, indicated by a less steep slope of the relationship of growth after versus before a disturbance. Even without knowing the ultimate cause of the growth decline episodes, especially for *P. dalatensis*, we can conclude that *P. kesiya* has a higher ecophysiological plasticity to adapt or acclimatise to lower-than-optimal site conditions than *P. dalatensis*, which responds more sensitively to environmental stress even in the "core" region of its occurrence. Altogether, our results on the growth and resilience of the two palaeotropical *Pinus* species are in good accordance with the differences in their current geographical distribution. Against the background of the decrease in the forest area of Vietnam and large-scale transformation of Vietnamese forests into grassland and

agricultural fields [65,66] and in the light of the small distribution range of *P. dalatensis*, measures to protect and conserve stands of this species should be perpetuated and further encouraged.

**Supplementary Materials:** The following material is available online at https://www.mdpi.com/article/10.3390/f12040511/s1, Figure S1. Chronologies of tree-ring stable carbon isotopes ($\delta^{13}$C) of two pine species. *Dashed lines* mean individual $\delta^{13}$C time series; *solid lines* mean $\delta^{13}$C time series. Figure S2: Correlation coefficients of $^{13}$C discrimination against atmospheric $CO_2$ ($\Delta^{13}$C) of the two pine species with *temperature* (*black bars*) and *precipitation* (*grey bars*) from 1901 to 2013. *Dotted lines* indicate thresholds of significance at $p < 0.05$. The letters on the *x*-axis indicate calendar months and seasonal periods. Table S1: Soil chemical properties of the plots with different *Pinus* species. *PIKE*: *Pinus kesiya*; *PIDA*: *Pinus dalatensis*.

**Author Contributions:** Conceptualisation, F.M.T. and L.T.H.; selection of study sites, F.M.T. and L.T.H.; wood and soil sampling, L.T.H.; dendrochronology and soil analyses, L.T.H.; analyses of stable isotope ratios, J.H.; data processing and evaluation, L.T.H. and J.H.; visualisation, L.T.H.; supervision, F.M.T.; funding acquisition, F.M.T. and L.T.H.; project administration, F.M.T. All authors contributed to writing and editing of all versions of the manuscript. All authors have read and agreed to the published version of the manuscript.

**Funding:** This work was supported by Bộ giáo dục và đào tạo through a PhD grant to L.T.H. (Grant No. 4358/QĐ-BGDĐT); and by the German Academic Exchange Service (Deutscher Akademischer Austauschdienst, DAAD; Project No. 57163751) through covering travel costs of F.M.T.

**Institutional Review Board Statement:** Not applicable.

**Informed Consent Statement:** Not applicable.

**Data Availability Statement:** All datasets presented in this study are included in the article/ Supplementary Materials.

**Acknowledgments:** The authors thank the head of Bidoup Nui Ba National Park, Le Van Huong, for the permission to conduct the study in the park, and the staff of the national park for their kind support. L.T.H. thanks Nguyen Thanh Tam for his assistance during fieldwork.

**Conflicts of Interest:** The authors declare no conflict of interest.

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
