# Peer review of "Resistance, Recovery and Resilience of Two Co-Occurring Palaeotropical Pinus Species Differing in the Sizes of Their Distribution Areas"

_forests, doi:10.3390/f12040511_

Round 1
Reviewer 1 Report
Dear Authors,
although I am not a dendroecologist, I think this is a good dendroecological study with application of state-of-the-art methodology and interesting results.
What I miss in the discussion is a critical reflection of the strengths and weaknesses especially of the RRR-method. There are some new articles on that (Bosse et al. 2020).
Moreover, the conclusions of the study could be improved. Last, but not least, there is need for a final language revision.
Best regards
Reviewer 2 Report
General comments:
Overview and general recommendation:
It is true that dendrochronology is a useful tool in determining long term trends in tree growth in relation to climate parameters such temperature and precipitation and a lot of effort has been put on this particular analysis. The aim of this article was to calculate resistant, recovery and resilience indices for two co-occurring pines regarding disturbances caused by weather extreme events.
The article's main contribution is that it gives detail information on tree ring dendrochronological methods, development of sensitivity indicators and cellulose extraction for stable carbon isotope analysis.
The Authors tried to present this method comprehensively which, I think it is a good idea. On one hand I found the manuscript overall well written and described. On the other hand I found some parts of the results section not so clear and straightforward presented. Therefore, I recommend that a minor revision is warranted. I explain below my concerns in more detail below and I ask the authors address my comments in their response.
Specific comments:
Lines 76-86
Usually, the last paragraph of the introduction section includes the aim and the scope of the study. I find this paragraph very general and I recommend to the Authors to state clearly and explicitly the aims of their study in order to be straightforward and clear to the reader
Lines 109-110
“… the litter is being annually burnt according to National Park rangers' common practice to foster tree 109 regeneration. “
Do you mean that prescribed burning it is officially for pine natural regeneration establishment? If yes then the more proper term “prescribed burning” must be used instead of “the litter is being annually burnt”
Lines 273-280
The authors here present results and numbers that are not connected with some figure or table. Are these results referred to figure 2?
Figure 3 and S1 captions
Mean seasonal periods is not enough, the Authors here must explain what the letters DJF, MAM etc mean. Additionally, there is no reference of these seasonal periods displayed in fig 3 in the results explaining what these parameters show and why they have been calculated for.
Line 322
“…showed generally higher Rc and Rs in PIKE compared to PIDA” As far as I can read from table 2 the Rc value in PIKE appears lower (1.48) than that of PIDA (1.51)
Lines 350-354
There must be a mistake or confusion regarding section 3.3 and fig. 5. I can’t figure out if these negative values are somehow correspond to fig 5, I don’t see a linear decrease for PIDA δ13C from 1901-2013. Please be more precise.
Lines 356-363
I can’t figure out how Rbar and EPS values presented here are connected with fig 5.
Line 392
As I was mentioned it before prescribed burning is more proper term for this regeneration method
